# Using Ensemble Diffusion to Estimate Uncertainty for End-to-End Autonomous Driving

Florian Wintel[*1], Sigmund Hennum Høeg[1], Gabriel Kiss[1], and Frank Lindseth[1]

[1]Norwegian University of Science and Technology
{florian.wintel, sigmund.hoeg, gabriel.kiss, frankl}@ntnu.no

## Abstract

End-to-end planning systems for autonomous driving are rapidly improving, especially in closed-loop simulation environments like CARLA. Many such driving systems either do not consider uncertainty as part of the plan itself or obtain it by using specialized representations that do not generalize. In this paper, we propose EnDfuser, an end-to-end driving system that uses a diffusion model as the trajectory planner. EnDfuser effectively leverages complex perception information like fused camera and LiDAR features, through combining attention pooling and trajectory planning into a single diffusion transformer module. Instead of committing to a single plan, EnDfuser produces a distribution of candidate trajectories (128 for our case) from a single perception frame through ensemble diffusion. By observing the full set of candidate trajectories, EnDfuser provides interpretability for uncertain, multimodal future trajectory spaces. Using this information we design a simplistic "safety-rule" that improves the system's driving score by 1.7% on the LAV benchmark. Our findings suggest that ensemble diffusion, used as a drop-in replacement for traditional point-estimate trajectory planning modules, can contribute to an uncertainty-aware decision making process in End-to-End driving policies by modeling the uncertainty of the posterior trajectory distribution.

## 1   Introduction

Uncertainty quantification (UQ) of machine learning systems is the problem of detecting situations in which a learned system cannot make a reliable prediction and is more likely to make a mistake [1]. UQ is especially important in the autonomous driving (AD) domain, where uncertainty about the correct action can have catastrophic consequences. Among other factors, the uncertainty of a learned AD system can be caused by sensor noise, wrong labels, real-world complexity, distribution shift, or architectural shortcomings [2]. Over the past decades, substantial effort has been dedicated to the estimation of uncertainty in learned systems, including Bayesian methods [3], Monte Carlo dropout [4], en-

sembles [5] and deterministic UQ methods [6]. In this work, we present a diffusion-based approach to uncertainty quantification.

Diffusion models are expressive generative models, proven to excel at modeling expressive distributions given data, like generating images [7, 8], video [9] and audio [10]. They are capable of modeling trajectories for motion planning and closed-loop robotic control tasks [11, 12], including autonomous driving [13–16]. A key to the success of diffusion models is that they can model multimodal distributions and are stable to train. In contrast to many traditional prediction models, which can only predict a single point estimate, diffusion models can generate an entire set of predictions for any single input.

In this study, we examine a diffusion model for end-to-end (E2E) autonomous driving. We approach uncertainty quantification through the introduction of a diffusion-based planner that can predict an arbitrary number of candidate trajectories in the closed-loop CARLA simulator [17]. Our method can assist in answering the following questions: When and where does the agent experience uncertainty, what is the cause, and what can it teach us about the underlying data distribution? Without changing the ground truth data or perception architecture of our baseline, we show that a probabilistic planner based on denoising diffusion can produce strong uncertainty estimates that can improve driving performance and provide insights into biases in the agent's training distribution. We demonstrate the potential benefit of uncertainty information for the end-to-end planning task by introducing a simple uncertainty-informed heuristic. Our work establishes a basis for advanced filtering strategies, capable of detecting uncertain, potentially dangerous situations in sparse driving data. Our contributions are as follows:

---

[*]Corresponding Author.

Proceedings of the 7th Northern Lights Deep Learning Conference (NLDL), PMLR 307, 2026.

- We present EnDfuser, a simple end-to-end driving agent capable of modeling planning uncertainty in closed-loop driving scenarios in the CARLA simulator.

- We show that a simple uncertainty-informed heuristic can increase the driving score of EnD-fuser by 1.7% in the LAV benchmark.

- We demonstrate that the posterior trajectory distribution can aid in extracting the long tail of the driving distribution by revealing occurrences of potentially safety-critical situations.

## 2 Related Work

### 2.1 UQ for closed-loop E2E AD

UQ is an essential aspect of autonomous driving systems, with research spanning across the domains of perception, prediction, planning, and control [2, 18]. Several studies have focused on UQ in closed-loop end-to-end planning approaches within the popular CARLA simulator [17]. Tai et al. [19] predict uncertainties over direct control actions. They choose a GAN-based approach in which the stochastic element is derived from a style transfer performed on the input image. Cai et al. [20] predict the variances of the speed and yaw distributions with a Gaussian mixture model (GMM). VTGNet [21] simultaneously predicts future trajectories, as well as the associated uncertainty of every trajectory position. More recently, VADv2 [22] models uncertainty implicitly by sampling from the planning action space in a probabilistic manner. It first defines a discretized action vocabulary of 4096 anchor trajectories and then assigns a probability to each candidate. Finally, TransFuser++ does not explicitly model uncertainty but has the ability to leverage the speed classifier's softmax confidence score in its control decision. However, this is limited to its prediction of longitudinal movement (velocity) and requires the use of a discrete speed classifier.

### 2.2 Diffusion models for AD and UQ

**Diffusion for AD planning.** Diffusion models [7] have been successfully applied to a wide range of perception tasks [23], as well as tasks in the domain of AD [24–26]. Several works on AD planning and control have adopted diffusion in their policies.

In the popular nuPlan simulator [27], Diffusion-ES uses unconditional diffusion to reduce the trajectory search space to the manifold of plausible trajectories w.r.t. the training set, then performs a gradient-free evolutionary search on the reduced solution space [13]. Diffusion Planner employs a conditional diffusion transformer [16]. In the non-reactive NAVSIM benchmark [28], DiffusionDrive extends a TransFuser [29] baseline with truncated diffusion on a set of noisy anchor trajectories, achieving real-time inference speed [14], while GoalFlow combines denoising diffusion and flow matching [30], using goal points for guidance [15]. In D4RL, a popular simulator for reinforcement learning (RL) agents [31], Venkatraman et al. adopt diffusion for their offline RL policy by producing latent candidates that are passed to a separate autoregressive policy decoder for direct action planning [32]. Likewise, Chu et al. integrate latent diffusion in their RL-based approach in the CARLA simulator [33].

**Diffusion for UQ.** Diffusion models have recently been proposed as a method for uncertainty modeling [34–36]. Shu et al. outline a UQ approach based on diffusion ensembles, which, in contrast to many other UQ methods, does not require UQ to be part of the model architecture [36]. Diffusion-based UQ has previously been applied to trajectory prediction [37–39]. In addition to these approaches, we leverage uncertainty information to increase the safety of our agent specifically in the AD domain. Although previous diffusion-based approaches in AD actively use the multimodal posterior distribution, they do not model uncertainty explicitly. To the best of our knowledge, this is the first work applying diffusion-based UQ for end-to-end imitation learning in closed-loop AD planning.

## 3 Method

### 3.1 Preliminaries

**End-to-end AD.** End-to-end AD is a motion planning and control task, in which a driving agent consumes raw sensor data and computes a motion plan or control action. In AD, the plan is often modeled as a trajectory in 2D space, where the ego vehicle is located at the coordinates $(0,0)$.

**TransFuser++.** We extend the TransFuser++ (TF++) agent [40]. TF++ achieves strong closed-loop performance in LAV, Longest6 and other end-to-end driving benchmarks in CARLA, and holds the second position on the CARLA leaderboard 2.0 [41]. TF++ is based on imitation learning (IL) and has a multitask architecture: Its perception encoder fuses visual information from RGB images with depth information from LiDAR bird's eye view (BEV) images. Plans are decoded by a transformer+GRU module that extracts waypoint queries, then processes them into spatial path coordinates and a target speed. An alternative variant, TF++ WP, only predicts a spatiotemporal trajectory. TF++ models speed prediction as classification and uses the softmax confidence score as a proxy for uncertainty [40]. EnDfuser adopts the perception module from TF++, but replaces all other modules with a simple probabilistic diffusion planner.

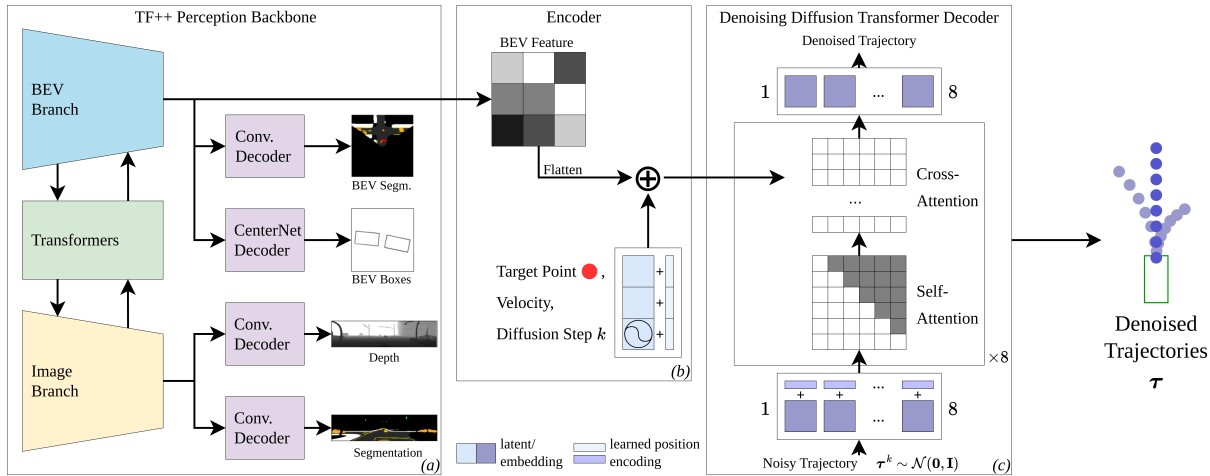

**Figure 1. EnDfuser architecture.** (a) The TransFuser++ perception backbone consumes two modalities, RGB images from the ego perspective and a LiDAR birds-eye-view (BEV) image. Transformer-based sensor fusion is performed between the two convolutional branches, after which four auxiliary perception tasks are learned (BEV segmentation, BEV object detection, ego perspective depth estimation and ego perspective segmentation). (b) We enrich the BEV features with a driving instruction target point (TP), the current velocity and the diffusion step $k$. (c) We iteratively denoise trajectories $\boldsymbol{\tau}^k$ sampled from a Gaussian, conditioning on the enriched BEV features via cross attention.

## 3.2 Planning with a diffusion model

We focus on UQ at the action level, specifically on the posterior action distribution predicted by a learned model. Diffusion models aim to model a distribution over a stochastic variable, given a data set $\mathcal{D} = \{\mathbf{x}_i\}_{i=1}^N$. When fitted to the data set, this allows us to retrieve samples distributed as the underlying data distribution $\tilde{\mathbf{x}} \sim p_\theta(\mathbf{x})$. In our AD application, we sample driving trajectories of the ego vehicle $\boldsymbol{\tau}$ given an observation $\mathbf{O}$.

We choose a denoising diffusion probabilistic model (DDPM) [7] as our underlying diffusion model. At the core of DDPM is the forward diffusion process, indexed with $k$, which adds noise to the sample from the data distribution $\boldsymbol{\tau}^0$ and ends in a known distribution like the Gaussian distribution.

$$\boldsymbol{\tau}^k = \sqrt{\bar{\alpha}_k}\boldsymbol{\tau}^0 + \sqrt{1-\bar{\alpha}_k}\epsilon, \quad \text{where } \epsilon \sim \mathcal{N}(\mathbf{0},\mathbf{I}). \quad (1)$$

The sequence $\bar{\alpha}_k$ is termed the *diffusion schedule* and corresponds to the amount of noise added to a sample at diffusion step $k$. A sample from a trained diffusion model is produced by an iterative process starting from a normally distributed value $\boldsymbol{\tau}^k \sim \mathcal{N}(\mathbf{0},\mathbf{I})$ and updated as

$$\boldsymbol{\tau}^{k-1} = \left(\gamma^k\boldsymbol{\tau}^k + \xi^k\boldsymbol{\tau}_\theta(\boldsymbol{\tau}^k,\mathbf{O},k)\right) + \Sigma_k\epsilon, \quad (2)$$

where $\boldsymbol{\tau}_\theta(\cdot)$ is the denoising network, $\epsilon \sim \mathcal{N}(\mathbf{0},\mathbf{I})$, and $\gamma^k, \xi^k$ are coefficients given by the diffusion schedule. The denoising network is trained to predict the clean sample given the corrupted sample, by minimizing

$$\mathbb{E}_{k,\boldsymbol{\tau}^0,\mathbf{O},\epsilon}\left[\left\|\boldsymbol{\tau}^0 - \boldsymbol{\tau}_\theta^0(\sqrt{\bar{\alpha}_k}\boldsymbol{\tau}^0 + \sqrt{1-\bar{\alpha}_k}\epsilon,\mathbf{O},k)\right\|^2\right]. \quad (3)$$

## 3.3 EnDfuser architecture

Parallel batch sampling is effectively realized using a GPU. The denoiser predicts a batch of trajectories $\mathcal{T}_t = \{\boldsymbol{\tau}_{t,i}\}_{i=1}^{N_{\text{infer}}}$, where $N_{\text{infer}}$ is the number of noisy input trajectories $\boldsymbol{\tau}^k$, and $t$ is the simulation frame. We condition the diffusion model on the perception input $\mathbf{O}_t$, which contains an RGB image, a LiDAR reading, a driving instruction target point (TP), and the current velocity. For inference, we adopt a DDIM schedule based on denoising diffusion implicit models (DDIM) [42]. DDIM is not bound to the Markovian process governing DDPM, which allows the model to sample from the target distribution using much fewer denoising steps. We obtain the perception encoding from the TF++ encoder's LiDAR bird's eye view (BEV) branch (Fig. 1(a)) and enrich the BEV encoding with embeddings of target point (driving instruction), velocity, and the diffusion step $k$ (Fig. 1(b)). Then we denoise the trajectory with a diffusion-based transformer decoder (Fig. 1(c)). Similarly to TransFuser, EnDfuser's output trajectory is represented as 8 waypoints spaced 250ms apart, always describing the plan for the next 2 seconds. The shape of the noisy trajectories $\boldsymbol{\tau}^k$ and the denoised trajectories $\boldsymbol{\tau}$ is $(8 \times 2)$, encoding 8 $(x,y)$ waypoint coordinates. We adopt the planning architecture introduced by Chi et al. [12], choosing a diffusion transformer decoder over a U-Net-based architecture. In the transformer block, each of the 8 noisy waypoints in $\boldsymbol{\tau}^k$ is represented with its own token embedding of size 256. This allows 8 waypoint queries to attend to the perception encoding (BEV feature memory) individually, effectively performing attention pooling as shown in Fig. 2.

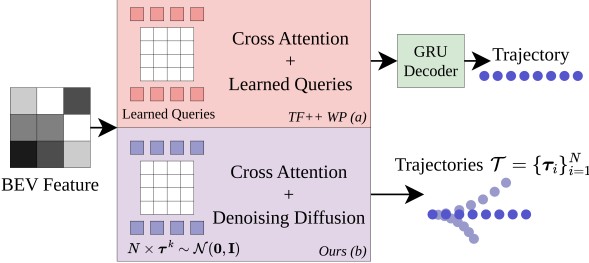

**Figure 2. Attention pooling on the BEV features.** (a) TF++ WP uses learned queries. (b) Our diffusion transformer uses noisy waypoint queries.

**Algorithm 1** Extracting uncertainties
**Data:** Observation $\mathbf{O}$
**Result:** Uncertainty estimates $\hat{\sigma}_s^2$, $\hat{\sigma}_y^2$
$\mathcal{T} \leftarrow \tau_\theta(\mathbf{O})$ `// sample a batch of trajectories`
**foreach** $\tau_i \in \mathcal{T}$ **do**
$\quad \tau_i \leftarrow \{wp_0, ..., wp_7\}$ `// 8 WPs, 250ms apart`
$\quad \kappa_i^{spd} \leftarrow \|wp_1 - wp_3\|^2 \times 2$ `// speed (m/s)`
$\quad wp\_dists \leftarrow \{d \in \|\tau_i\|^2 : d \le maxdist\}$
$\quad aim\_idx \leftarrow \arg\max(wp\_dists)$
$\quad \kappa_i^{yaw} \leftarrow \arctan2(wp_{aim\_idx}) \cdot \frac{180°}{\pi}$ `// yaw (deg.)`
**end**
$\hat{\sigma}_s^2 \leftarrow \hat{\sigma}^2(\{\kappa_i^{spd}\})$, $\hat{\sigma}_y^2 \leftarrow \hat{\sigma}^2(\{\kappa_i^{yaw}\})$

## 3.4 Measuring uncertainty

EnDfuser's output representation $\mathcal{T}_t$ comprises 8 $(x, y)$ waypoint coordinates, i.e. 16 variables. To reduce the complexity of interpreting $N$ candidates, we transform $\mathcal{T}_t$ into a bivariate set of *control commands* $\mathcal{K}_t$. We extend the PID control logic of TF++ WP [40] for the transformation, as it directly pertains to the driving task. The desired speed and yaw angle are first calculated based on the predicted trajectory, before they are transformed into acceleration, braking, and steering commands. We apply these operations to all trajectories in $\mathcal{T}_t$. The extraction logic is described in Alg. 1 (omitting frame $t$ for brevity). Speed is determined by the Euclidean distance of two waypoints. Yaw is inferred from the angle between the ego vehicle's origin $(0, 0)$ and a dedicated aim waypoint "aim_idx", where "wp_dists" are the distances between any waypoint and the origin, and "maxdist" is the maximum allowed distance of "aim_idx" from the origin. We adopt the fixed values "$wp_1$","$wp_3$" and "maxdist" from the PID controller of TF++ WP [40]. Like in previous works [20], we can model two uncertainties, speed and yaw. We define $\hat{\sigma}^2(\mathcal{K}_t^{spd})$ and $\hat{\sigma}^2(\mathcal{K}_t^{yaw})$, respectively.

In practice, we rely only on the speed uncertainty measure for two reasons: First, the calculated yaw is not independent of speed, since the choice of "aim_idx" depends on the waypoint distances. Second, Jaeger et al. identify speed as the main source of multimodality in the task design of CARLA leaderboard 1.0, since the route to follow is defined unambiguously by the target points [40]. In this work we therefore opt for the speed variance $\hat{\sigma}^2(\mathcal{K}_t^{spd})$ as our primary uncertainty indicator and refer to it as $\hat{\sigma}_s^2$ for brevity. Instances of high $\hat{\sigma}_s^2$ are of particular interest to us. To emphasize the correlation of $\hat{\sigma}_s^2$ with safety-critical events, we implement an optional rule-based safety system. The added safety rule states that the agent should override the desired speed with a value of $0\,\text{m/s}$ if $\hat{\sigma}_s^2$ exceeds a given threshold $\lambda$, forcing the agent to brake. We find that this simple addition marginally improves EnDfuser's infraction score, as seen in Section 4.

## 3.5 Implementation

We train EnDfuser with imitation learning using the publicly available TransFuser++ data set [40, 43], which was recorded by an expert demonstrator. The expert is a rule-based agent that can access privileged information from the CARLA simulator, such as the locations of the ego vehicle and obstacles [29, 40]). This privileged information is unavailable to the sensor-based EnDfuser. Ground truth data are collected by having the expert traverse the training towns. The training samples used by EnDfuser include recorded observations $\mathbf{O}$ (LiDAR, RGB image, speed, and next target point (TP)) and are labeled with the expert's driven trajectories $\tau_{gt}$ (a set of 8 2D points from the ego vehicle's frame of reference). The TPs are GNSS-based anchor points on the town maps (30 meters apart on average) and describe the route to follow. The full TF++ dataset has some $555,000$ training samples. TF++ uses the expert's path (lateral plan) and target speed (longitudinal plan) instead of trajectories. This is required to model speed multimodally. EnDfuser does not require this path+speed split since it can model the multimodal trajectory distribution directly. Models are evaluated in a closed-loop manner by running the agent through evaluation routes in the CARLA simulator. The observed metrics are driving score (DS), route completion (RC) and infraction score (IS), where RC is the average route completion percentage, IS is a geometric series of infraction penalties for collisions and red-light infractions in $(0, 1)$ and DS is the weighted sum of every per-route RC multiplied by the per-route IS [29].

**Training.** We train EnDfuser with DDPM and 100 denoising steps. The diffusion model can be trained to predict noise $\epsilon$, or trajectories $\tau$. Although both training approaches produce functional driving policies, we find that a trajectory prediction network $\tau_\theta$ can denoise a valid trajectory within only 2 DDIM steps, while a noise prediction network $\epsilon_\theta$ requires at least 10 DDIM steps to produce an equivalent level of driving proficiency. As computational requirements scale linearly with the number of

**Table 1. Runtime scaling of EnDfuser**: EnDfuser with $N_\tau = 128$ and 2 DDIM steps runs only slightly slower than TF++, but provides a rich set of candidate trajectories.

| Steps | $N_\tau$ | time (ms) ↓ | time (FPS) ↑ |
|---|---|---|---|
| 2 | 128 | 0.0335 | 29.047 |
| 4 | 128 | 0.0386 | 25.881 |
| 8 | 128 | 0.0477 | 20.940 |
| 16 | 128 | 0.0662 | 15.116 |
| 2 | 1 | 0.0344 | 29.787 |
| 2 | 256 | 0.0374 | 26.742 |
| *TransFuser++* | | *0.0300* | *33.252* |

diffusion steps, we opt for the trajectory prediction network. The models are trained on 2 A100 GPUs with a batch size of 48. Training follows the general regime of TF++ and is performed in two stages: First, the TF++ perception backbone is pre-trained for 31 epochs on the 4 perception tasks shown in Fig. 1(a). Then the full EnDfuser architecture is trained end-to-end for an additional 61 epochs.

**Inference.** We evaluate using a DDIM schedule with 2 steps, after which we choose a single candidate trajectory to follow. As the sequential denoising process introduces additional computational overhead, keeping the number of denoising steps low helps maximize inference speed. This does not apply to the number of sampled candidates $N_\tau$: Sampling from the noise prior is trivial and $N_\tau$ is only limited by the GPU's parallel processing capability. We tested this on an NVIDIA RTX 4090 GPU with different configurations. Table 1 shows that the inference speed does not scale considerably with the number of simultaneously predicted trajectory candidates, only with the number of applied denoising steps. Using 2 DDIM steps, EnDfuser can produce up to 128 candidate trajectories simultaneously before any substantial slowdown occurs. The resulting framerate of 29.047 FPS is only marginally slower than TransFuser++. To achieve real-time performance, we choose $N = 128$ for our further experiments.

# 4 Experiments and Results

## 4.1 Experiment setup

We evaluate our agent on the LAV [44] and Longest6 [29] benchmarks in CARLA. These were created as local alternatives to the official CARLA leaderboard 1.0, which relies on external servers. In the benchmarks, agents must navigate scenarios from the NHTSA pre-crash scenario typology [45], with simulated traffic, weather and daylight conditions, as well as predefined adversarial scenarios.

Longest6 combines the 6 longest routes from CARLA towns 1 to 6 for a total of 36 routes with an average length of 1.5km. It is considered a training benchmark, i.e. the agents are evaluated on the same 36 routes in the same 6 towns which also constitute the training environment. In contrast, the shorter LAV benchmark excludes towns 2 and 5 from the training data, then uses only them as the evaluation environment. LAV is nevertheless easier to solve than Longest6, due to the higher traffic density present in the latter. To account for the stochastic nature of the evaluations, we train using 3 different seeds for each agent configuration, then evaluate each model 9 times and present the average score.

## 4.2 Speed variance and safety rule

We evaluate different EnDfuser variants, exploring the effect of variance threshold $\lambda$ on closed-loop driving performance. Tables 2 and 3 show the top-scoring EnDfuser configurations. On the LAV benchmark, EnDfuser's base configuration outperforms TF++ in DS and IS. With the safety rule, it achieves an additional reduction of 0.05 in vehicle collisions per kilometer (compare Table 2). Although all EnDfuser variants perform less well overall on Longest6 than TF++, the safety rule still decreases the overall vehicle collision rate from 1.1 to 1.02 collisions per kilometer on this benchmark. However, this does not result in a higher DS because the gain in IS is offset by a reduction in RC due to an increase in agent timeouts (compare Table 3). We find that $\lambda$ values between 0.3 and 0.4 yield the highest scores. A lower $\lambda$ is detrimental to RC and a higher $\lambda$ shows lower increases in IS. On LAV, $\lambda = 0.4$ yields the highest improvement, without reducing RC.

**Relevance of speed variance.** While the results with the active safety rule differ only marginally from the EnDfuser baseline, there is a clear difference in overall driving behavior. The average agent speed decreases with lower $\hat{\sigma}_s^2$ thresholds $\lambda$, as the agent brakes more often. EnDfuser's average speeds are higher in LAV (8.169 km/h) than Longest6 (5.308 km/h), indicating a higher tolerance for delays. This is likely due to higher traffic density in Longest6, and coincides with a lower timeout rate (compare Tables 2 and 3). This presents the possibility that any improved IS on LAV is a consequence of lower average speeds, rather than braking intelligently. We test this by reducing EnDfuser's speed naïvely by 0.5 km/h and 1.0 km/h, to observe the effect of different average speed reductions on LAV. While Fig. 3 shows that a larger speed reduction achieves a similar increase in DS as the safety rule, it coincides with increased route timeouts. We conclude that the safety rule is more likely to brake when the agent enters potentially dangerous situations.

**Table 2. LAV evaluation.** The metrics are driving score (DS), route completion (RC) infraction score (IS), collisions with pedestrians, vehicles and static objects, red lights, stop signs, route deviations, timeouts and the agent becoming blocked. Average scores and standard deviations are determined over 27 LAV evaluations (3 model seeds and 9 repetitions). The highest non-expert scores are printed in **bold**, second highest are underlined.

| Agent | DS↑ | RC↑ | IS↑ | Ped↓ | Veh↓ | Stat↓ | Red↓ | Dev↓ | Stop↓ | TO↓ | Block↓ |
|---|---|---|---|---|---|---|---|---|---|---|---|
| Ours, no rule | 76.4 ±5 | 98.7 ±2 | 0.773 ±0.05 | **0.00** | 0.37 | 0.06 | 0.12 | **0.14** | 0.00 | **0.04** | 0.01 |
| Ours, $\lambda = 0.4$ | **78.1** ±4 | 98.9 ±1 | **0.789** ±0.05 | **0.00** | **0.32** | 0.03 | 0.13 | 0.16 | 0.00 | 0.07 | 0.01 |
| Ours, $\lambda = 0.3$ | 77.1 ±5 | 98.7 ±1 | 0.781 ±0.05 | **0.00** | 0.34 | 0.06 | 0.14 | 0.15 | 0.00 | 0.08 | 0.01 |
| TF++ [40] | 70 | **99** | 0.70 | 0.01 | 0.63 | **0.01** | **0.04** | 0.26 | 0.00 | 0.05 | **0.00** |
| *Expert* [40] | *94* | *95* | *0.99* | *0.00* | *0.02* | *0.00* | *0.02* | *0.00* | *0.00* | *0.00* | *0.08* |

**Table 3. Longest6 evaluation.** The metrics are driving score (DS), route completion (RC), infraction score (IS), collisions with pedestrians, vehicles and static objects, red lights, route deviations, timeouts and the agent becoming blocked. Average scores and standard deviations are determined over 27 Longest6 evaluations (3 model seeds and 9 repetitions). The highest non-expert scores are printed in **bold**, second highest are underlined.

| Agent | DS↑ | RC↑ | IS↑ | Ped↓ | Veh↓ | Stat↓ | Red↓ | Dev↓ | TO↓ | Block↓ |
|---|---|---|---|---|---|---|---|---|---|---|
| Ours, no rule | 62.6 ±6 | 91.8 ±3 | 0.669 ±0.06 | 0.01 | 1.10 | 0.05 | 0.11 | 0.01 | 0.16 | **0.05** |
| Ours, $\lambda = 0.4$ | 61.9 ±6 | 90.1 ±3 | 0.673 ±0.06 | 0.01 | 1.02 | 0.03 | 0.10 | **0.00** | 0.20 | **0.05** |
| Ours, $\lambda = 0.3$ | 61.6 ±6 | 91.6 ±3 | 0.656 ±0.07 | 0.01 | 1.11 | 0.04 | 0.10 | **0.00** | 0.17 | **0.05** |
| TF++ [40] | **69** | **94** | **0.72** | **0.00** | **0.83** | **0.01** | **0.05** | **0.00** | **0.07** | 0.06 |
| *Expert* [40] | *81* | *90* | *0.91* | *0.01* | *0.21* | *0.00* | *0.01* | *0.00* | *0.07* | *0.09* |

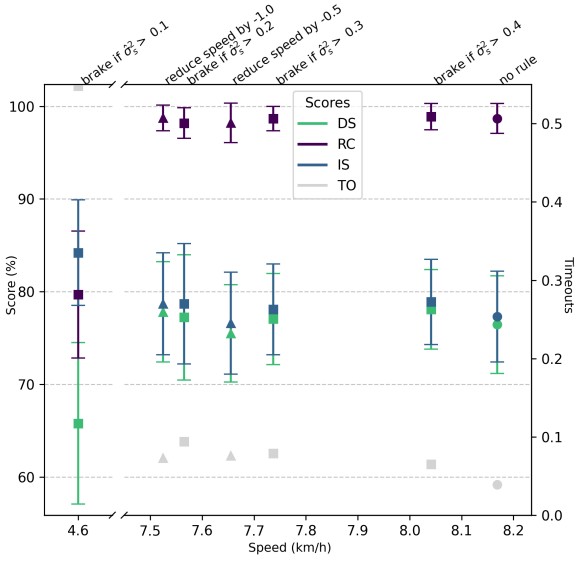

**Figure 3. Effect of different driving rules in LAV.** We compare three different agent types: Baseline EnDfuser ○, EnDfuser + naïve speed reduction (km/h) △ and EnDfuser + safety rule □.

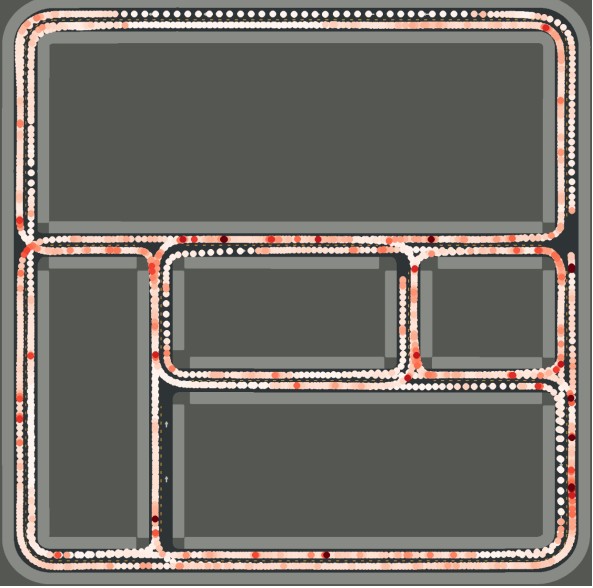

**Figure 4. Uncertainty map in Town02.** Desired speed variances for 6 routes driven by EnDfuser, downsampled to 2Hz and color coded from $\hat{\sigma}_s^2 = 0$ (○) to $\hat{\sigma}_s^2 = 0.6$ (●). Elevated variance is visible at intersections and bends. All towns can be found in the appendix.

## 4.3 Uncertainty map

In the following experiments, we use $\hat{\sigma}_s^2$ to detect high-uncertainty events. We collect $\hat{\sigma}^2(\mathcal{K}_t^{spd})$ at evaluation time for each inference frame $t$. We track the agent's speed variance and locations at any given point along the evaluation routes for a full Longest6 evaluation (approximately $700,000$ frames). Instances of $\hat{\sigma}_s^2 > 0.4$ appear in fewer than $0.001\%$ of all frames. We then localize uncertainty regions by observing the agent's positions where high variance was recorded. Figure 4 associates

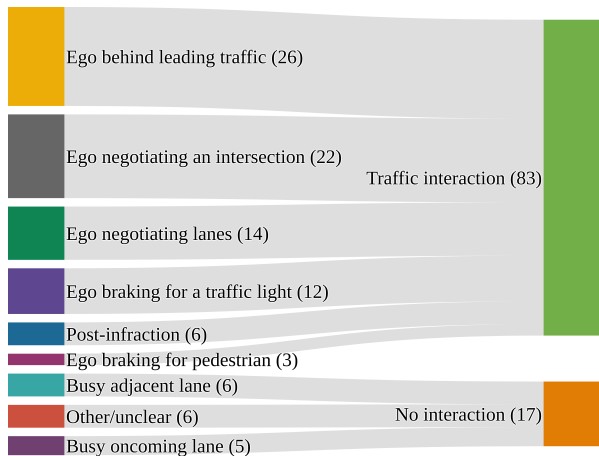

**Figure 5. Categories of uncertain situations.** The majority of uncertainty spikes coincides directly with traffic interactions. We investigate the agent's context in the 100 least certain situations by recording the sensory input of one Longest6 evaluation (36 episodes) and extracting the 100 frame sequences with the highest variance values $\hat{\sigma}^2(\mathcal{K}_t^{spd})$.

the variances with the points along the route where they occurred. There are clear clusters of uncertainty near intersections and bends, where the ego vehicle is more likely to interact with other traffic, than on straight stretches of road. Occurrences of high variance ("spikes") coincide with the locations of generated adversarial scenarios in the CARLA routes. This implies that speed uncertainty can be used to pinpoint high-risk events and possibly to filter for challenging segments in training data sets.

## 4.4 Categorization

For an informed visual inspection of uncertain situations, we record the sensor readings and plan output for one full Longest6 evaluation and extract the frame sequences around the 100 highest uncertainty values. We then categorize the circumstances surrounding the spikes. As shown in Fig. 5, most uncertain situations occur during interactions with other agents. Of the 100 events, 83 occur during agent interactions, with 36 being highly dynamic ones, in which the agent changes lanes or crosses junctions, such as the example in Fig. 6(a). This coincides with the two most common infractions in EnDfuser and TF++, i.e., invading occupied lanes and not yielding to other traffic at intersections (the latter case being mentioned as one of TF++'s failure modes on the CARLA leaderboard 2.0 [41]). In 17 of the 100 observed cases, no clear source of uncertainty is discernible through visual inspection. We interpret such behavior as instances of causal confusion. EnDfuser either slows down during the spike or, if already standing still, experiences the spike before accelerating. As Longest6 is a training benchmark,

it is also possible that the model associates static scene elements with driving behaviors. Collisions occur in 7 of the 100 inspected scenes, while unsafe behavior (cutting traffic, halting without reason) appears in 9 additional cases, Figure 6(b) illustrates the moment before one such collision during a lane change. Note that the randomly selected speed (in magenta) resulted in suboptimal behavior here. Had the safety rule been active, the agent could have overridden the suboptimal speed prediction with 0 m/s and forced the ego vehicle to brake.

## 4.5 Further observations

**Multimodality.** The training objective of the diffusion model is to predict a representative sample of the ground truth trajectory distribution (as discussed in Section 3.2), allowing it to capture multimodality. Multiple modes are sometimes apparent in the desired speed distributions, like the one seen in Fig. 6(b). This implies that the posterior speed distribution contains more granular uncertainty information than can be captured by a simple variance-based measure and that more sophisticated measures (e.g., entropy) may use it more effectively.

**Aleatoric uncertainty.** The unpredictable movement of other agents, as well as traffic signals, appears to be linked to high speed uncertainty, like in Fig. 6(a). We interpret this, at least partially, as an expression of aleatoric uncertainty, which is inherent to the environment and cannot be reduced by adding more driving demonstrations during training.

**Lateral label noise.** Through further empirical observation, we discovered that another source of uncertainty is label noise in the training data. This uncertainty pertains to lateral movement rather than speed. Like TF++, EnDfuser always receives the next TP along the route as its driving command, but no instruction beyond this. In Fig. 6(c), the planned trajectory extends beyond the known TP. Such occurrences introduce high lateral uncertainty in the predicted plan trajectories $\mathcal{T}$, indicating strong lateral conditioning on the target point and suggesting the presence of data noise in the training setup and expert data. Incidentally, this occurs far enough from the vehicle's origin to be filtered out by the transformation $\mathcal{T} \rightarrow \mathcal{K}$ in Alg. 1, which only considers a short planning horizon and discards information further than 1 second into the future (speed) or more than 3 meters away (yaw). Choosing a different transformation operation could cause erratic driving behavior. The observation may also offer an explanation why using two consecutive TPs did not result in improved driving in recent work [41].

## 4.6 Limitations

EnDfuser fails to predict some safety-critical situations, possibly due to insufficient, one-sided coverage

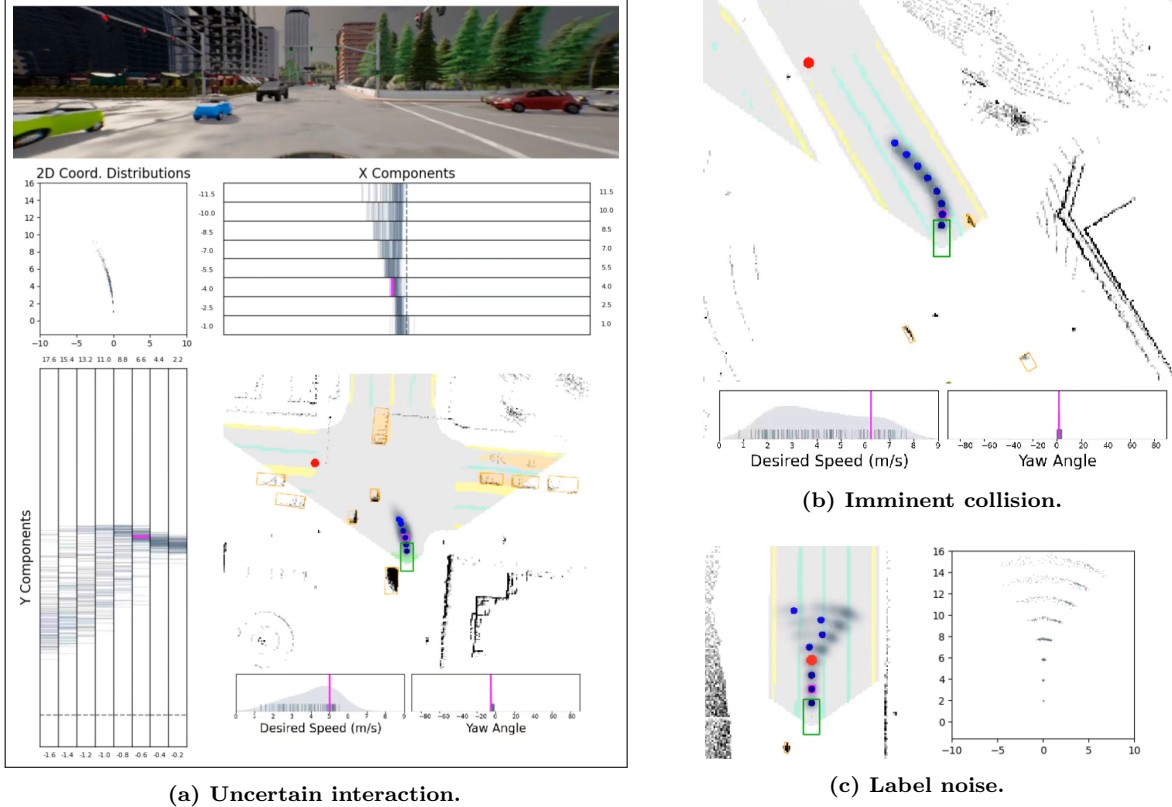

**Figure 6. Instances of trajectory disagreement in $\mathcal{T}$.** Desired speed and yaw angle represent $\mathcal{K}_t$ (KDE with Scott's rule for visualization). The selected action is marked in magenta. (a) Most instances of high variance are interactions with dynamic objects like other agents. (b) Elevated longitudinal trajectory disagreement is observed *before* a collision. (c) The prediction horizon extends beyond the target point, forcing the agent to predict positions for which it has no driving instruction. This results in lateral trajectory disagreement.

during training. An example can be found in appendix A.1. Our experiments also reveal significant noise in the key metrics, with $\sigma$(DS) up to 6%, and DS ranging from 56.8% to 67.7% in Longest6. This limits comparability, given the considerable computational resources required by even a modest 27 repetitions per experiment. As a consequence, the EnDfuser baseline is yet to be compared with established non-diffusion UQ methods like GMM. Future work should explore more difficult settings like the CARLA leaderboard 2.0/2.1 or real-world settings, using a more informed uncertainty measure. Although speed variance is an easily controllable quantifier, it may not be sufficient for interpreting larger speed ranges and scene complexities. In addition to speed, uncertainty measures should consider yaw, which requires disentangling its representation from speed. Furthermore, variance alone cannot distinguish between aleatoric and epistemic uncertainty. For instance, some out-of-distribution frames (e.g. post-infraction frames) can be visually identified (there are no infractions in the training data), but this does not replace a quantitative distinction. Possible candidate measures include entropy, as well as density-based measures like mode count and curvature. Finally, the braking heuristic based on a hard-coded threshold is simplistic and not expected to generalize. Future research should explore dynamic approaches as well as learned safety heuristics.

## 5   Conclusion

We introduced EnDfuser, a simple yet powerful AD motion planning model based on denoising diffusion and show its efficacy on the LAV and Longest6 benchmarks. Using the diffusion policy, we achieve effective, real-time uncertainty modeling by generating a set of 128 candidate trajectories simultaneously. By modeling the variance of the predicted speed distribution, we demonstrate that this set captures the model's prediction uncertainty and can be incorporated into the agent's planning process. The resulting 1.7% increase in driving score in LAV and our extensive visual investigation highlight the potential for more sophisticated heuristics informed by the posterior trajectory distribution $\mathcal{T}$. Our ensemble diffusion method can also be used to extract areas of high agent uncertainty at test time, including instances with label noise, possibly facilitating data set mining by filtering for the long tail of the driving distribution.

# Acknowledgements

This research received funding from the PERSEUS project, a European Union's Horizon 2020 research and innovation program under the Marie Skłodowska-Curie grant agreement No 101034240. The authors acknowledge the financial support of MoST (MobilitetsLab Stor-Trondheim, https://www.mobilitetslabstortrondheim.no/en/).

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

# A   Appendix

## A.1   Limitation example

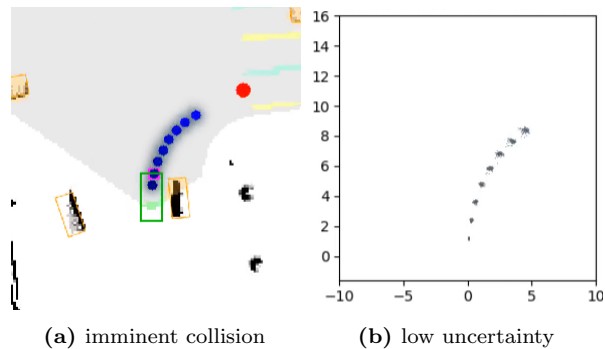

**(a)** imminent collision   **(b)** low uncertainty

**Figure A.1.   Prediction failure.**   (a) EnDfuser
changes lanes while taking a right turn. It ignores the
vehicle to its right and causes a collision. (b) No spike
in uncertainty is detectable before the collision.

## A.2   Diffusion steps

Inference speed scales linearly with the denoising
schedule. As a consequence, we want to use as few
denoising steps as possible. Increasing the number
above 2 can yield higher driving scores, but not
sufficiently to justify the slower inference speed.

**Table A.1. Ablation study.** Longer denoising sched-
ules have a stronger effect on inference speed than on
driving performance, as demonstrated on LAV.

| Steps | FPS ↑ | DS ↑ | RC ↑ | IS ↑ |
|---|---|---|---|---|
| 2 | 29.047 | 76.4 ±5 | 98.7 ±2 | 0.773 ±0.05 |
| 4 | 25.881 | 78.1 ±4 | 99.0 ±1 | 0.790 ±0.04 |
| 8 | 20.940 | 76.3 ±6 | 98.3 ±2 | 0.776 ±0.06 |
| 16 | 15.116 | 77.8 ±6 | 98.2 ±1 | 0.793 ±0.06 |

## A.3   Uncertainty maps

We record the speed variances for a full Longest6
evaluation. Areas with a regular occurrence of ele-
vated speed variance are clearly visible around inter-
sections and bends. Each town displays the variances
of 6 cumulative routes driven by EnDfuser, down-
sampled to 2Hz and color coded from $\hat{\sigma}_s^2 = 0$ (○) to
$\hat{\sigma}_s^2 = 0.6$ (●) in the speed predictions. Town06 in par-
ticular has long stretches with elevated uncertainty.
All towns can be inspected below.

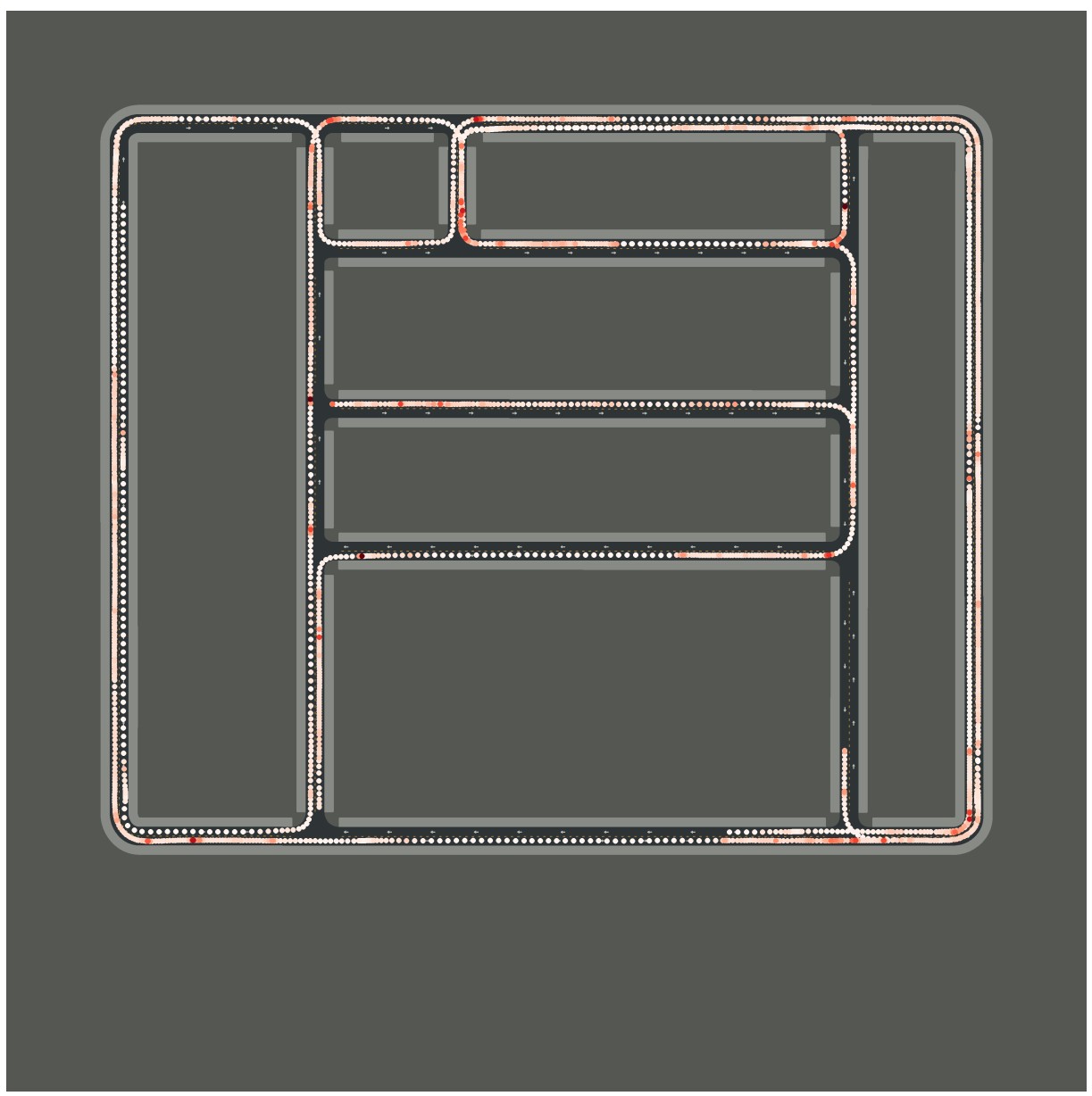

**Figure A.2. Uncertainty map in Town01.**

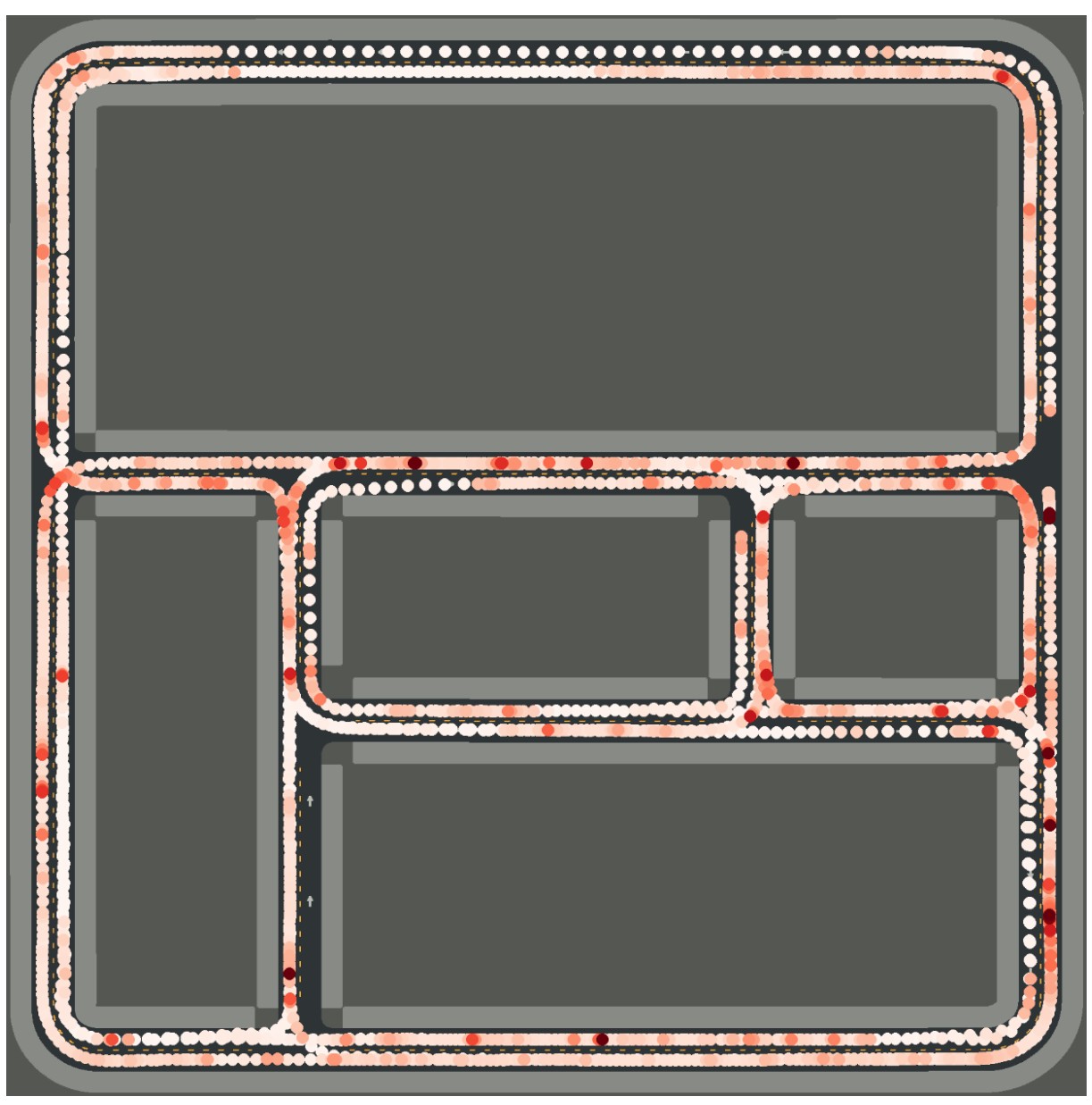

**Figure A.3. Uncertainty map in Town02.**

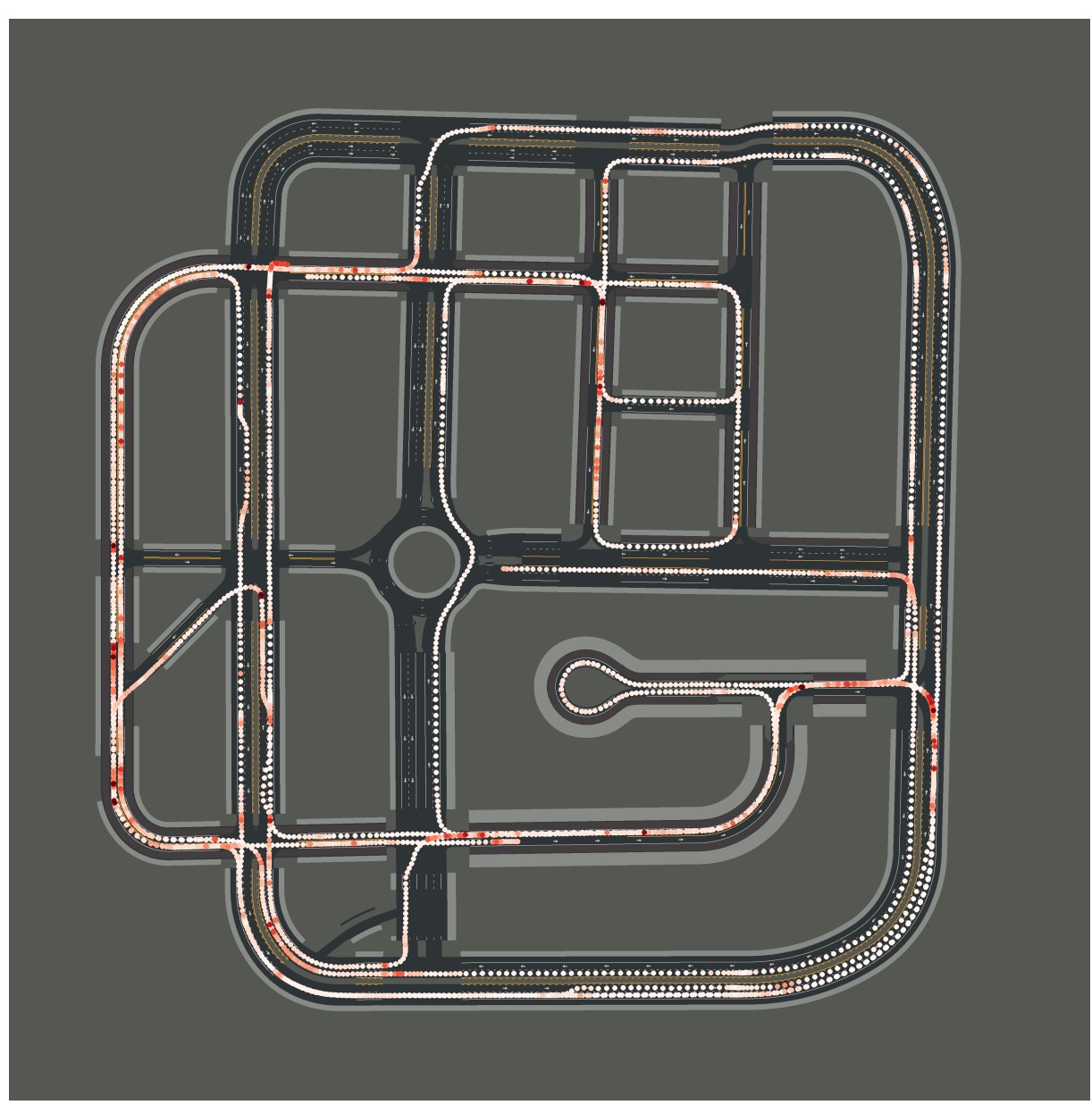

**Figure A.4. Uncertainty map in Town03.**

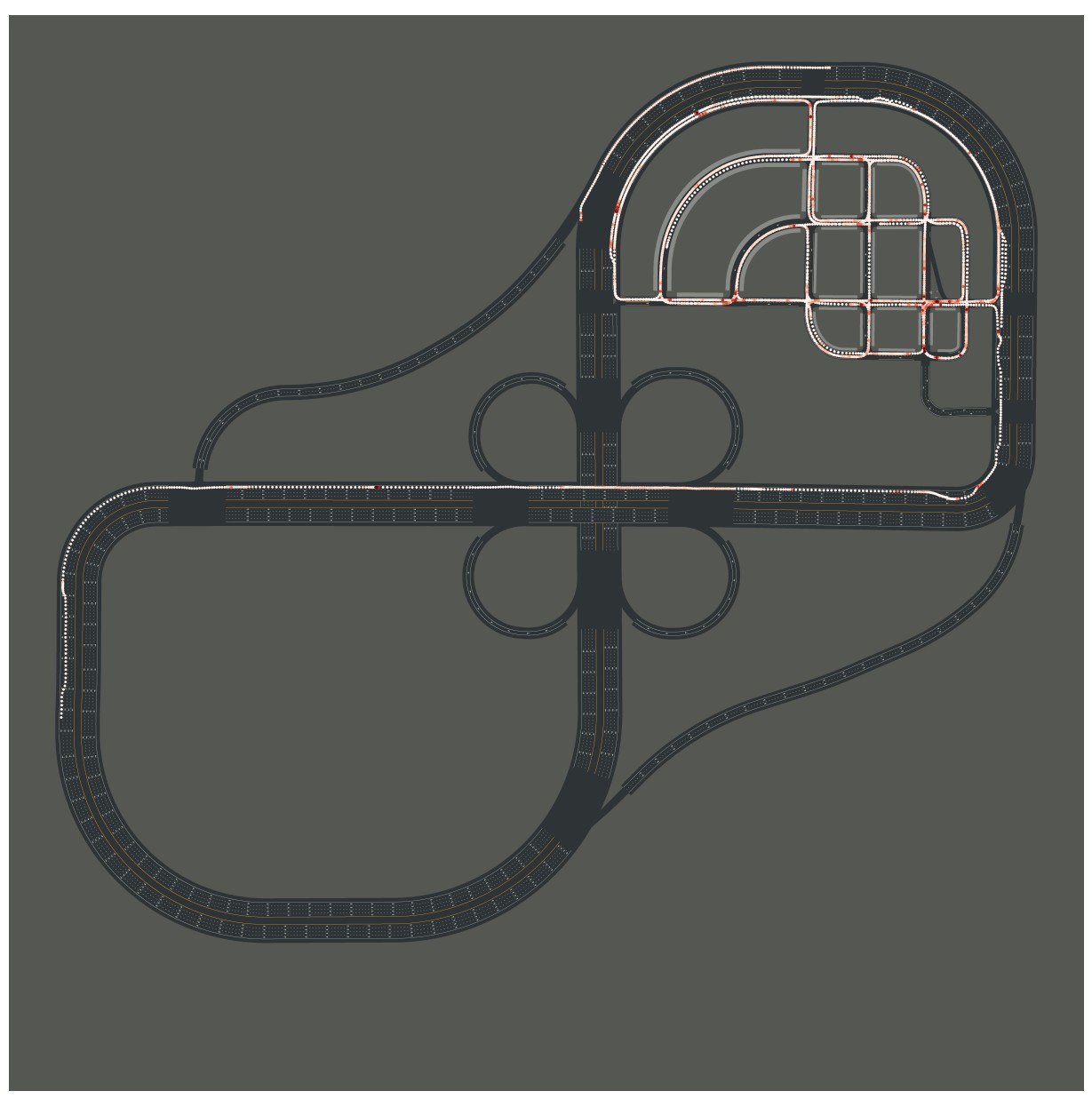

**Figure A.5.** Uncertainty map in Town04.

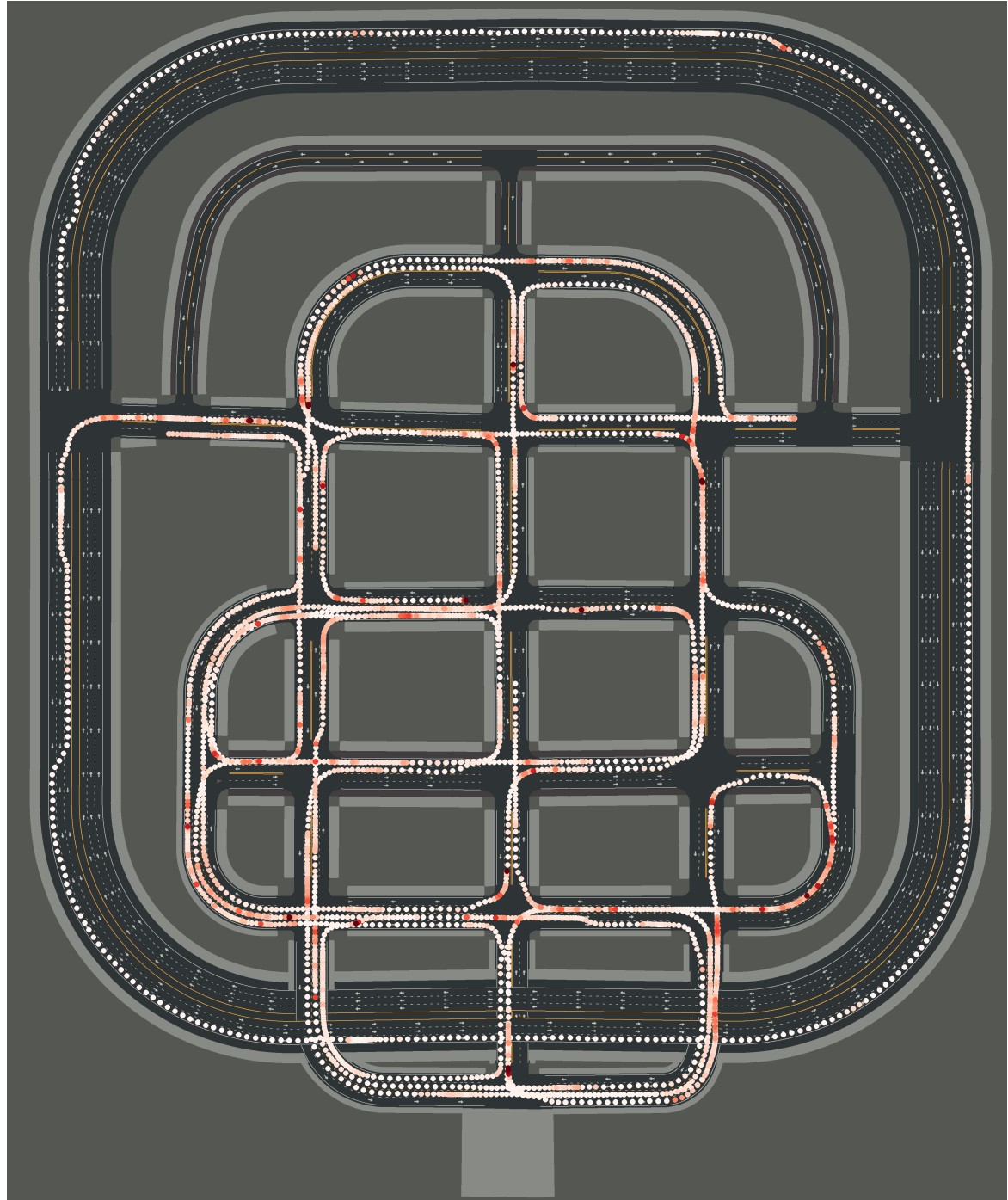

Figure A.6. Uncertainty map in Town05.

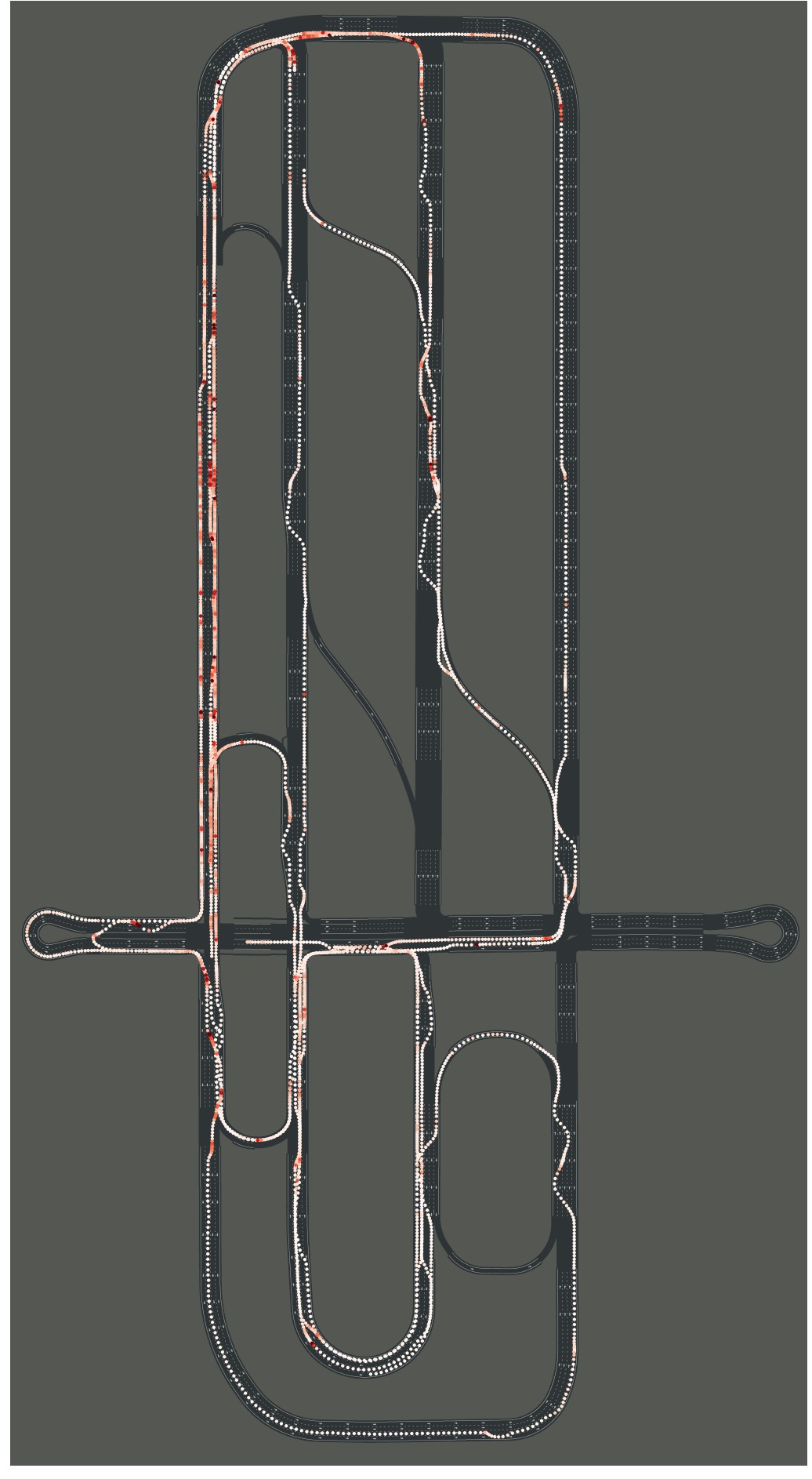

Figure A.7. Uncertainty map in Town06.

