# OpenReview forum: "Using Ensemble Diffusion to Estimate Uncertainty for End-to-End Autonomous Driving"
_NLDL.org/2026/Conference — NLDL 2026 Oral_

### Official Review · Reviewer_eJaC · 2025-09-22
**A good paper with undue and immodest claims**

**Rating:** 4
**Confidence:** 4
**Final Rating:** 4
**Final Confidence:** 4

**Summary:**

This paper introduces EnDfuser, an end-to-end driving agent that integrates diffusion models into trajectory planning for autonomous driving. The main contribution is the use of ensemble diffusion to generate multiple candidate trajectories simultaneously, thus enabling a form of uncertainty quantification (UQ). The paper evaluates EnDfuser in the CARLA simulator, comparing it with the strong TransFuser++ baseline. Results show that EnDfuser improves the driving score on the LAV benchmark by +1.7%, mainly due to the use of speed variance as a proxy for uncertainty and the introduction of a simple safety rule that brakes the vehicle when variance exceeds a threshold. Additional analyses include uncertainty maps, qualitative categorization of high-uncertainty events, and discussion of aleatoric vs. epistemic uncertainty.

Overall, the paper is well-written, clear, and provides interesting insights into the intersection of diffusion models and uncertainty modeling for autonomous driving. While the methodology is sound and the experiments are correctly executed, the claims about safety improvements may be somewhat overstated, given the modest quantitative gains, reliance on a single uncertainty measure, and limited evaluation setting.

**Strengths:**

-The integration of diffusion-based generative models with UQ in autonomous driving is timely and innovative. It addresses a gap in the literature where most E2E planners focus on single-point trajectory predictions.

-By framing UQ in terms of multiple candidate trajectories, the paper provides a useful perspective on how generative models can be leveraged for safety in AD. The uncertainty maps are particularly illustrative, highlighting correlations between high-uncertainty regions and traffic interactions.

-The authors provide a thorough evaluation on two benchmarks, reporting not only aggregate metrics but also breakdowns of infractions and qualitative cases. They also control for randomness through multiple seeds and repetitions.

-The idea of using ensemble diffusion as a “drop-in replacement” for traditional planners could inspire further research, particularly if extended to stronger benchmarks (e.g., CARLA leaderboard) or real-world driving.

**Weaknesses:**

-The exclusive reliance on speed variance as the uncertainty proxy is limiting. Many critical driving scenarios involve lateral decisions or multimodal futures that speed variance alone cannot capture.

-The method does not differentiate epistemic uncertainty (due to model limitations) from aleatoric uncertainty (due to environment stochasticity). This distinction is important for diagnosing failures and improving safety.

-The heuristic of braking whenever uncertainty exceeds a threshold, while illustrative, is context-insensitive and may create new risks (e.g., unnecessary braking leading to rear-end collisions).

-The reported improvement of +1.7% in driving score seems to be modest. Moreover, the authors acknowledge high variance in results (standard deviation up to 6%), which questions the robustness of the claimed gains.

-All experiments are in CARLA, and some benchmarks (e.g., Longest6) overlap with training data. Evaluation on CARLA leaderboard 2.0/2.1 or real-world data would provide stronger evidence.

-The paper suggests that ensemble diffusion increases safety overall, but this seems overstated given the modest results, failure to anticipate some critical situations, and dependence on a single metric.

Questions to the authors:

1) Have you considered using entropy or multimodal density measures as alternative UQ indicators?

2) How do you ensure that improvements are not merely due to reduced average speed rather than genuinely better handling of uncertainty?

3) Could the method be extended to incorporate adaptive or learned safety rules instead of fixed heuristics?

**Final Justification:**

The authors agreed to moderate their claims and offered more reasonable justifications for the system's design.

**Justification:**

This paper makes a solid and interesting contribution to the field of autonomous driving by exploring ensemble diffusion as a trajectory planner that naturally provides uncertainty estimates. The methodology is technically correct, the writing is clear, and the experiments are executed carefully. However, several limitations reduce the strength of the conclusions: the reliance on a single and simplified uncertainty measure, the limited and partly overlapping benchmarks, the modest and unstable performance improvements, and the somewhat strong claims regarding safety.

In spite of these weaknesses, the paper is valuable and original, and it opens a promising line of research at the intersection of diffusion models and uncertainty quantification. My recommendation is accept (4/5), with the caveat that the authors should moderate their claims, clarify limitations, and possibly extend the evaluation in future work.

---

> ### Author Rebuttal · Authors · 2025-10-21
>
> We thank the reviewer for their comprehensive and insightful review. We are encouraged that they agree with our research direction, presentation and methodology, and we appreciate their constructive feedback.
>
> - __W1, W2, W3__: We agree with these points. We acknowledge the need for future research to include all degrees of freedom into the uncertainty estimate, to model uncertainties multimodally, to distinguish between types of uncertainty, and to finally utilize the extracted uncertainty information in a manner that generalizes.
> - __W5__: We agree that this is a limitation.
> - __W4, W6__: We understand the need to clarify our claims accordingly.
>
> - __Q1 - Multimodal measures__: We suspect that there is a relationship between the multimodality in the trajectory distribution and the nature of uncertainty present (aleatoric vs epistemic).
> We have performed limited experiments with density-based methods (KDE, histograms) to factor in mode count and mode curvature, as well as compound measures using more than one variable. However, our results have thus far been inconclusive.
> - __Q2 - Impact of average speed vs. uncertainty handling__: The uncertainty maps in Fig. 4 and in the appendix show qualitatively that the EnDfuser agent is not just slower on average but is more likely to slow down in specific areas. These areas coincide with denser traffic, and with more infractions. The observed improvement on LAV could still be a result of lower average speed in such locations, but likely not global average speed alone. This also stresses the need for future investigation in environments with a higher speed range (e.g.CARLA Leaderboard 2.0/2.1).
> - __Q3 - Adaptive or learned safety rules__: There has been some investigation into adaptive heuristics outside the diffusion domain, such as the uncertainty-based action fusion in PMP-net [1]. Based loosely on the concept, one could have two policies, one probabilistic default-policy (e.g. trajectory-based EnDfuser), and one deterministic fallback (e.g. path+speed-based TF++), with an uncertainty-based tradeoff between the two policies. On the other hand, we would expect *learned* safety rules to require either a shift to closed-loop training regimes (like reinforcement learning), or expert demonstrators that can provide uncertainty labels.
>
> __Claims, limitations and future work__: We understand the need for moderation and clarification and will amend the manuscript as requested.
> We will moderate the claims regarding performance and safety, and clarify that the core contribution lies in obtaining diffusion-based uncertainty information and using it for decision making and interpretability.
> In addition, the manuscript will be expanded with respect to limitations and future work, e.g. replacing the limited, threshold-based braking heuristic with more dynamic approaches, as well as investigating multimodality-aware uncertainty measures like entropy.
>
> ---
> [1] P. Cai, S. Wang, Y. Sun, and M. Liu, “Probabilistic End-to-End Vehicle Navigation in Complex Dynamic Environments With Multimodal Sensor Fusion”, IEEE Robotics and Automation Letters, vol. 5, no. 3, pp. 4218–4224, July 2020, doi: 10.1109/LRA.2020.2994027.

---

### Official Review · Reviewer_7BS6 · 2025-10-07
**Review for submission #22**

**Rating:** 4
**Confidence:** 4
**Final Rating:** 4
**Final Confidence:** 4

**Summary:**

This paper presents EnDfuser, an autonomous driving system that uses a diffusion model to generate an ensemble of 128 candidate trajectories instead of a single one. The authors measure the model's uncertainty by calculating the speed variance across this ensemble.  They demonstrate the car brakes when uncertainty is high. This rule improves the overall driving score by 1.7% on the LAV benchmark.

**Strengths:**

1. The authors apply diffusion-based uncertainty quantification for an end-to-end imitation learning agent in a closed-loop driving setting, which is novel to me.
2. There is a safety benefit. The simple uncertainty-informed rule improves the driving score by 1.7% and reduces vehicle collisions on the LAV benchmark.
3. The method is efficient as it archieves 30 FPS, which is important to real-time deployment.

**Weaknesses:**

1. The authors use speed variance to capture the uncertainty. Would it be helpful if more information is considered?
2. The rule is designed in the specific dataset. Would it be general when we shift from other scenarios?

**Final Justification:**

After reading the rebuttal, most of the previous concerns are addressed. I recommend acceptance at this stage.

**Justification:**

The framework is interesting and the auhtors demonstrate its real-world application. I also have some questions about the system design. I hope the authors can discuss it.

---

> ### Author Rebuttal · Authors · 2025-10-21
>
> We thank the reviewer for their constructive feedback and for pointing out the novelty, potential safety benefit and efficiency of our method. We would be happy to discuss any questions:
>
> 1. In future research, we plan to extract more information from the posterior trajectory distribution than speed variance. For instance, driving performance is lacking when the agent is required to perform lane changes in dense traffic (most evident in Town06). Thus, disentangling yaw from speed information, and modeling yaw uncertainty in addition to just speed, could alleviate this. In addition, other measures like entropy (as pointed out in one of the other reviews), or density could reflect the multimodality of the posterior trajectory distribution better than mere variance and give rise to more sophisticated safety heuristics.
>
> 2. The evaluations we use for the speed rule (Longest6 and LAV) contain a subset of the scenarios in the NHTSA pre-crash typology (common pre-crash conditions), but these are explicitly simulated. Considering the rule's simplistic design, it would be interesting to explore how well it generalizes to real-world environments.
> Due to the PID-based control logic of the TF++ baseline, the speed rule is aimed at an environment with a concept of speed as a distinct, controllable variable. However, the earlier TransFuser agent has been deployed in several data/simulation environments. EnDfuser’s speed-based braking rule could be directly transferred to other CARLA-based environments, but it may need to be adapted for other settings like the real-world-based NAVSIM environment.
>
> We acknowledge the potential for future improvements and will amend the paper accordingly.

---

### Official Review · Reviewer_Lwn5 · 2025-10-08
**Using Ensemble Diffusion to Estimate Uncertainty for End-to-End Autonomous Driving**

**Rating:** 4
**Confidence:** 3

**Summary:**

This paper presents a method for using uncertainty produced by diffusion models to achieve safer autonomous driving. The general idea of the paper is to produce many different plans through the denoising process. From these trajectories we can then compute a variance on the plans and use that variance the estimate the uncertainty of the model. The paper focuses specifically on the variance of the speed in the plan (as it is difficult to decouple yaw from speed). They implement a hard-coded rule to break when the uncertainty of the model (computed like above) is above some threshold. The idea is to simply break when the model is uncertain as that is a safer action. They find their method with the heuristic is able to perform better on the LAV evaluation than the bustling they build on and worse on the Longest6 evaluation in comparison to the baseline.

**Strengths:**

High relevance. Uncertainty quantification is an important research direction for deploying ML in the real world. Without an accurate sense of confidence, deploying ML models will always be risky. As a result, this is a highly relevant topic to research. As well, self-driving is an important and cutting edge research direction so that is also important. I think they are tackling an important problem here.

Good presentation. The paper is clear to read with good figures and includes information like confidence intervals.

The work seems to be correct. They provide a reasonable set of seeds to verify their results and provide a description of their method that I believe is enough to reimplement if required.

Overall, this paper is well written and tackles an interesting problem. I was able to easily understand their method, and believe it is a reasonable idea and research direction. I think this work also provides interesting future research directions as there are many similar and advanced problems.

**Weaknesses:**

The largest weakness of this paper is the mediocre performance of the method on the Longest6 evolution. Instead of performing better than the work they base their paper on it performs a good amount worse. While this is a weakness, I don't believe that the paper should be punished for it. Negative results are still results, and I think it instead provides an interesting future direction to work on.

I think the experiments are slightly limited as well. The authors mention there are some harder driving domains they want to try their method on but I see no reason other that it takes time that they wouldn't try them here. The results they present seem reasonable but could likely be expanded without too much effort.

**Justification:**

This paper works on an interesting and difficult problem (uncertainty in autonomous driving), provides a reasonable solution and provides reasonable results. I think this paper should be accepted since there is nothing wrong with the paper as far as I can tell. If I had to give complaints, I would say the results are not the most impressive and the experiments seem somewhat limited. I think that both of these are ok though as the first gives rise to future work and the second is just because the domain is more difficult to test than other domains.

---

> ### Author Rebuttal · Authors · 2025-10-21
>
> We thank the reviewer for their helpful feedback and are encouraged that they agree with the relevance, presentation, and implementation of our research.
> 1. We agree with the weaknesses that were pointed out. We suspect that the lower performance on Longest6 compared to the baseline can be partially attributed to not modeling yaw uncertainty, which we believe to be important, for example for navigating the dense stop-and-go traffic in Town06.
> 2. We agree that this is a limitation. Future research should aim to evaluate the agents in the mentioned harder driving domains, to establish their ability to generalize.
>
> Both points warrant further investigation, and we will amend the limitations section of the paper to reflect this.

---

### Official Review · Reviewer_zCbr · 2025-10-08

**Rating:** 4
**Confidence:** 2
**Final Rating:** 4
**Final Confidence:** 4

**Summary:**

The paper introduces EnDfuser, an end-to-end autonomous driving system that uses an ensemble diffusion model to estimate uncertainty in its planning. Instead of predicting a single trajectory, EnDfuser generates a distribution of 128 possible trajectories from sensor data. By measuring the variance in speed across these candidate trajectories, the system quantifies its uncertainty. The authors demonstrate that applying a simple safety rule (braking when this speed variance exceeds a threshold) improves the driving score by 1.7% on the LAV benchmark by reducing collisions. This approach not only enhances safety but also helps identify and interpret challenging, high-uncertainty driving scenarios.

**Strengths:**

- This paper presents a novel use of diffusion models for uncertainty quantification in the complex, high-stakes domain of end-to-end autonomous driving. It claims to be the first work to apply diffusion-based UQ for imitation learning in a closed-loop AD planning context.

- The research provides concrete evidence that its uncertainty-aware approach enhances safety. By implementing a simple safety rule based on speed variance, the EnDfuser agent improves its driving score by 1.7% on the LAV benchmark, primarily by reducing vehicle collisions.

- A notable strength is that the method is computationally efficient. It can generate a large ensemble of 128 candidate trajectories with only a marginal impact on inference speed, running at nearly 30 FPS, which is crucial for practical application in autonomous vehicles.

- EnDfuser is proposed as a "drop-in replacement" for traditional deterministic planners. This modularity makes the concept practical and suggests it could be integrated into various existing autonomous driving architectures to improve their safety and robustness.

**Weaknesses:**

- The paper's main practical demonstration hinges on a rule that forces the vehicle to brake if speed variance exceeds a threshold. This is a blunt instrument that lacks nuance. It doesn't differentiate between a truly dangerous situation (e.g., a pedestrian stepping out) and a benign but complex one (e.g., a four-way stop with hesitant drivers). This could lead to overly cautious or even dangerous behavior, such as unnecessary emergency braking that could cause a rear-end collision.

- The headline 1.7% improvement in Driving Score on the LAV benchmark is statistically weak, especially given the high standard deviation of up to 6% reported in the results. More importantly, on the more challenging Longest6 benchmark, the safety rule actually decreased the overall Driving Score. The paper's own experiment (Figure 3) shows that a naive, constant speed reduction achieves a similar performance boost, which undermines the claim that the uncertainty-based braking is uniquely "intelligent."

- The entire method relies almost exclusively on the variance of predicted speed. The authors admit that yaw (lateral movement) is also a source of uncertainty but dismiss it due to its entanglement with speed. This is a major shortcoming, as lateral uncertainty (e.g., swerving into another lane) is arguably more critical for safety than longitudinal uncertainty. The method fails to capture a crucial dimension of planning risk.

- The paper introduces a diffusion-based UQ method but fails to compare it against other established UQ techniques like Monte Carlo dropout or deep ensembles within the same autonomous driving framework. It only compares EnDfuser to the deterministic TransFuser++ baseline. Without this comparison, it's impossible to know if ensemble diffusion is actually a superior approach for UQ in this context or just one of many ways to model uncertainty.

**Final Justification:**

The primary strength of this work is its novel and timely application of ensemble diffusion for uncertainty quantification (UQ) in the challenging domain of end-to-end autonomous driving. The authors convincingly demonstrate that this approach is not merely theoretical; it is computationally efficient enough for real-time application and provides a tangible, interpretable method for improving safety.

The work also has limitations. The core "safety rule" is admittedly simplistic, the performance gains are modest, and the reliance on speed variance alone as an uncertainty metric is a clear simplification of a more complex problem.

However, I feel the core contribution is still worth sharing, so I lean towards acceptance.

**Justification:**

The primary strength of this work is its novel and timely application of ensemble diffusion for uncertainty quantification (UQ) in the challenging domain of end-to-end autonomous driving. The authors convincingly demonstrate that this approach is not merely theoretical; it is computationally efficient enough for real-time application and provides a tangible, interpretable method for improving safety.

The work also has limitations. The core "safety rule" is admittedly simplistic, the performance gains are modest, and the reliance on speed variance alone as an uncertainty metric is a clear simplification of a more complex problem.

However, I feel the core contribution is still worth sharing, so I lean towards acceptance.

---

> ### Author Rebuttal · Authors · 2025-10-21
>
> We thank the reviewer for their valuable feedback. We are encouraged that they see the potential of using a diffusion-based trajectory planning module to achieve uncertainty quantification with high computational efficiency. We will happily address their concerns below.
>
> __Simplistic speed rule__: This is an important limitation of the speed heuristic. A speed rule with a hard-coded threshold is simplistic and likely does not generalize well. Future research should explore more sophisticated (and possibly adaptive) safety rules.
> __Lack of improvement__: We agree that the quantitative improvement based on the simple speed-based braking rule is statistically weak. However, the uncertainty maps in Fig. 4 and in the appendix show some qualitative evidence that the agent does not just slow down on average but is more likely to do so in specific areas. These areas coincide with denser traffic, and with more infractions.
> __Yaw uncertainty__: We agree with the reviewer, and we suspect that not modeling yaw uncertainty contributes to the observed lack of performance, especially on Town06 in Longest6, which contains challenging lane-changes in dense stop-and-go traffic.
> __UQ baselines__: We acknowledge the necessity of comparing the diffusion-based approach to UQ baselines. Future work in this direction should consider the quality of the uncertainty estimates, as well as the computational demand to obtain them.
>
> We will update our manuscript to more clearly state the limitations, and how they may be addressed in future research.

---

### Meta-Review · Area_Chair_HvWg · 2025-10-28

**Recommendation:** Accept (Oral)
**Confidence:** 4

**Metareview:**

This paper investigates end-to-end autonomous driving. And important yet often ignoring issue is uncertainty, as uncertainty provides valuable information for planning and interpretability. This paper proposes to quantify uncertainty through diffusion models. Results show that this approach not only improves driving performance, but gives further insights and handling the inherent long tail distribution of driving scenarios. All reviewers agree that the problem is relevant, the approach is interesting, and the overall contribution is clear. The AC agrees with the unanimous view of the reviewers and follows their feedback for acceptance.

---

### Decision · Program_Chairs · 2025-11-05

**Decision:**

Accept (Oral)

**Comment:**

We recommend an oral and a poster presentation given the AC and reviewers recommendations.